# Nanoparticles Based on Novel Carbohydrate-Functionalized Polymers

**DOI:** 10.3390/molecules25071744

**Published:** 2020-04-10

**Authors:** Cláudia D. Raposo, Cristiano A. Conceição, M. Teresa Barros

**Affiliations:** LAQV-REQUIMTE, Department of Chemistry, NOVA School of Science and Technology, Universidade NOVA de Lisboa, 2829-516 Caparica, Portugal; piccfa@gmail.com (C.D.R.); ca.conceicao@campus.fct.unl.pt (C.A.C.)

**Keywords:** PLGA, PEG, polymeric nanoparticles, glucose, galactose, thymidine, mannose, coumarins, drug delivery systems

## Abstract

Polymeric nanoparticles can be used for drug delivery systems in healthcare. For this purpose poly(lactic-co-glycolic acid) (PLGA) and poly(ethylene glycol) (PEG) offer an excellent polymeric matrix. In this work, PLGA and PEG polymers were functionalized with coumarin and carbohydrate moieties such as thymidine, glucose, galactose, and mannose that have high biological specificities. Using a single oil in water emulsion methodology, functionalized PLGA nanoparticles were prepared having a smooth surface and sizes ranging between 114–289 nm, a low polydispersity index and a zeta potential from −28.2 to −56.0 mV. However, for the corresponding PEG derivatives the polymers obtained were produced in the form of films due to the small size of the hydrophobic core.

## 1. Introduction

Described as the manipulation of atomic matter, nanotechnology was described theoretically in the 1960s by Richard Feynman, and the practice emerged a decade later. After Taniguchi’s, Drexler’s, and other scientist’s valuable contributions, nanomedicine has developed [1,2] and recently, the three main applications of nanomedicine are in tissue engineering, nanoprobes, and nanoparticles for drug delivery.

Suitable nanoparticles are capable of transporting drugs in a targeted manner to a specific tissue, cell or organ, minimizing the toxic effects and high therapeutic doses inherent in most current pharmacological treatments [3,4,5]. Recent advances in drug encapsulation and/or delivery of nanoparticles have demonstrated the enormous potential that these nanomaterials can have in healthcare, and their ability to improve the pharmacokinetic and pharmacodynamic properties of an active ingredient, thereby increasing the effectiveness of treatment and reducing the toxicity to patients [6].

Depending on the application (diagnosis, imaging, or therapy), different types of nanoparticles have been proposed, and these can be divided into two main groups: organic and inorganic nanoparticles. In the first group are dendrimers, liposomes, and polymeric nanoparticles. The second group includes quantum dots, silica, gold, and silver nanoparticles. Due to the high rate of tissue accumulation, that can result in toxicity problems, nanoparticles that cannot be degraded by the body are not as popular and attractive as biodegradable and biocompatible polymeric nanoparticles (PNPs) [7,8].

PNPs play an important role in therapeutic applications, such as encapsulation and controlled drug release [9,10]. The PNP matrix, size [11,12,13], surface area [14], shape [15,16], and surface electrical potential [17,18,19,20] are important to circumvent the limitations inherent in drug delivery and must be appropriate for the intended application.

Targeted drug delivery, also known as “magic bullet”, is a scientific concept proposed by Paul Ehrlich in 1900, and is the ability to selectively eliminate pathogenic microorganisms without harming the patient themselves, intending that the drug only acts on its intended target. Unlike most currently available treatments, PNPs capable of targeting drug delivery to a particular tissue, cell or organ can be achieved if the surface of the PNP is functionalized accordingly (derivatization with antibodies or their modified fragments, proteins, peptides, oligonucleotides, or carbohydrates) [19,20,21]. Highly specific interactions occur between these ligands and certain receptors of different cell types. Lectins, for example, present in different organs such as liver cells or over-expressed on the membrane surface of cancer cells, are able to specifically internalize monosaccharide molecules such as lactose, glucose, galactose, mannose, amongst others, suggesting the ability to target or accumulate PNPs in these types of lectin-containing cells [21,22].

With all of this in mind, we have attempted to functionalize commercial polymers, namely poly(ethylene glycol) (PEG) and poly(lactic-co-glycolic acid) (PLGA), in order to obtain drug delivery agents, either pro-drugs or nanoparticles for drug encapsulation. Monosaccharides (glucose, galactose, mannose, and thymidine) were bonded to these polymers to allow specific interactions between cell receptors and drug delivery agents. Coumarins were also included to potentially increase the pharmaceutical activity of these polymers, since they are known for their biological properties [23,24,25,26], but also to exploit their use as traceable probes due to their fluorescence character. In some cases, the click chemistry methodology [27] was used to establish the covalent bonds between these building blocks. More information about the formation of triazole rings with polymers can be read in an extensive review written by Arslan and co-workers [28], and in a complete review of polymeric scaffolds for tissue engineering, written by Zou and co-workers [29].

## 2. Results and Discussion

Our synthetic strategy began with the preparation of ligands that would be bonded to poly(ethylene glycol) (PEG) and poly(lactic-co-glycolic acid) (PLGA) polymers. We wanted these bonds to be strong, but also easily formed, and both 1,2,3-triazole ring and amide linkages present those features, whereby coumarins, carbohydrates and the polymers were modified accordingly, as described below.

### 2.1. Ligand Synthesis and Preparation—Coumarins

Coumarins can be obtained commercially or synthesized either by Pechmann condensation [30], Knoevenagel condensation [31], Perkin reaction [32], or Reformatsky reaction [33], the first two methods being the most widely used. For this work, however, only one coumarin was synthesized. Pechmann reaction between 2-methylresorcinol and diethyl-2-acetylglutarate in acidic media afforded **1** in 60% yield (Scheme 1).

Propargylation reactions of **1**, and other commercially available coumarins, namely umbelliferone, 3-carboxylic acid coumarin, and 4-methyl-7-hydroxy coumarin, are presented in Scheme 2. The propargylation conditions used were the ones already reported in the literature [34,35]. Alkyne coumarins **2**, **4**, **5**, and **6** were obtained in excellent yields, and pure enough for further reactions. NMR and FT-IR spectra (see Appendix A for ^1^H, ^13^C and MS/MALDI-TOF spectra for novel synthesized compounds) were in accordance with those already reported in the literature. For propargylated coumarin **4**, methylene protons were observed at 4.77 ppm, and the alkyne proton at 2.58 ppm. Alkyne stretch bands were observed at 3266 and 2119 cm^−1^. Coumarin **5** presented the methylene proton signal at 4.95 ppm and the alkyne proton at 2.56 ppm. FT-IR spectrum presented the alkyne stretches at 3315 and 2117 cm^−1^. Ester hydrolysis of **2** was accomplished with sodium hydroxide in ethanol at 80 °C, affording **3** in 95% yield.

In some cases, a linker was necessary in order to bond the coumarin, or the carbohydrate, moiety to the polymer and for these cases compound **7** was prepared. Due to its high volatility and reactivity, **7** was always maintained in diethyl ether, even for its quantification and characterization. Subsequent triazole ring formation with coumarin **6,** afforded **8** in good yield (Scheme 3).

### 2.2. Ligand Preparations—Carbohydrates

Specific carbohydrate derivatization was challenging due to the presence of several hydroxyl groups having similar reactivities. However, depending on the reaction conditions, the anomeric hydroxyl can be preferential replaced. Furthermore, in order for a carbohydrate to be recognized by lectins, it often must be non-reducing.

Insertion of a triazole ring between the carbohydrate moiety and the coumarin/polymer moieties not only increases the linkage strength, but mimics the amide group, which can induce, or improve, the biological activity of the final compounds. Propargylation of glucose, galactose, and mannose (Scheme 4) using immobilized sulfuric acid on silica [36], afforded the necessary terminal alkynes, **9**–**11**, for posterior cycloaddition reactions. The propargylation yields obtained were poor to moderate (38%–68%), and the products were mixtures of α and β anomers, with a 2:1 ratio. This ratio was calculated taking advantage of the vicinal coupling constants characteristics for hexoses (α proton near 5 ppm as a broad singlet and β proton around 4.50 ppm, with ^3^*J*_H1–H2_ = 7.9 Hz), and with ^1^*J*_H-C_ coupling constant for mannopyranose **11**. Benzylation of α and β propargyl glucopyranosyl mixtures afforded the per-*O*-benzylated anomers, which could be separated by column chromatography, using 6:1 hexane/ethyl acetate as eluent.

Acetylation of propargyl glucopyranose **9**, afforded **12**. Subsequent triazole formation with azide **7** afforded **13** in good yield. De-acetylation using sodium methoxide led to the corresponding product **14** (Scheme 5).

### 2.3. Ligand Preparations—Thymidine

Tosylation of the primary hydroxyl group of 2’-deoxythymidine was accomplished using tosyl chloride (**15**, Scheme 6). Subsequent reaction with sodium azide, either by conventional heating in dimethylformamide (DMF) using an oil bath (85 °C, 15 h, 87%), or using microwave irradiation (100 °C, 250 W, 2 min, 90%), afforded the corresponding azide **16**. Although a yield improvement was not significant using microwave heating, the reaction time was drastically reduced.

Having the ligands already prepared, they were used to prepare PEG and PLGA derivatives, as described in the following sections.

### 2.4. PEG Derivatives

PEG is a versatile polymer for pro-drug development due to its high solubility in water and available commercially at various molecular weights, and its stealth properties [37]. PEG pro-drugs often diminish collateral effects and can be specifically administered on targeted tissues [38]. Although the conjugation of pharmaceutical agents with PEG has brought significant improvements such as an increased solubility in water, prolonged liberation rates, and reduction of toxicity, problems such as the necessity for high doses and drug resistance are still awaiting resolution. Multi-therapy presents itself as a viable solution since polymeric pro-drugs are able to form nanoparticles that are able to encapsulate a different drug, allowing the reduction of administered doses, and consequently an increased therapeutic efficiency [39,40,41]. 

Esterification of coumarin 3-carboxylic acid with PEG using *N*,*N*-dicyclohexylcarbodiimide (DCC), and catalytic 4-dimethylaminepyridine (DMAP, Scheme 7) afforded the fluorescent polymer **17** in excellent yield. Product formation was observed by FT-IR, which confirmed the presence of the ester linkage at 1766 cm^−1^. The proton NMR spectrum showed the presence of one coumarin moiety, determined by comparing the relative intensity of the coumarin signals to PEG signals, since PEG with an average molecular weight of 1000 g/mol was expected to present between 84 and 96 protons. Integration of aromatic signals between 8.57 and 7.32 ppm for one coumarin, resulted in 88 PEG protons (between 4.60 and 3.50 ppm), which is within the expected values. This is a very important result, since the other PEG hydroxyl should be free to link to the carbohydrate moiety. A ^13^C NMR spectrum supports the esterification success through coumarin signals between 117 and 163 ppm. The terminal hydroxyl of ester **17** was then tosylated to provide ester **18**, in good yield. A tolyl methyl proton signal was observed at 2.45 ppm and a corresponding carbon signal at 21.6 ppm. Aromatic proton signals where observed between 7.90 and 7.32 ppm, and between 136.5 and 115.5 ppm in the carbon spectrum. The tosyl group was substituted using sodium azide in DMF, affording the azide **19** in very good yield. Product formation was confirmed by the absence of tosyl signals both in the proton and carbon spectra, and by the observation of an azide stretch band in the FT-IR at 2104 cm^−1^. Cycloaddition with the previously prepared propargyl galactopyranose **10** using catalytic CuI, and diisopropylethylamine (DIPEA) in freshly distilled tetrahydrofuran (THF) afforded the desired triazole-containing glycoconjugate **20** in only 18% yield. The poor yield obtained is believed to be due to copper complexation with the free hydroxyls of propargyl galactopyranose. Adding 2,2’-bipyridyl in excess to the reaction mixture, did not improve this result [42]. A Proton NMR spectrum showed the triazole proton signals shifted between 8.23 and 7.99 ppm, and the coumarin signals in the expected aromatic region, namely between 8.60 and 7.30 ppm. Galactopyranose signals were observed between 5.10 and 4.40 ppm, with PEG signals between 4.30 and 3.34 ppm. Carbon spectrum confirmed these results, mainly by the presence of the anomeric carbon at 98.8 ppm and triazole signal at 124.9 ppm. Matrix-Assisted Laser Desorption/Ionization with Time-of-Flight analyser (MALDI-TOF) mass spectrum presented a medium mass/charge of 1512, where the expected value was 1401 *m*/*z*., which result is within the range for (C_2_H_4_O)_n_H_2_O, with n = 21–24.

In order to form symmetric or asymmetric PEG derivatives diazide **22** was prepared through the sodium azide reaction with ditosylate **21** (Scheme 8). Formation of di-tosylated product **21** was confirmed by proton NMR, where the relative intensity of the tosyl signals, compared to the PEG methylene signals, were as expected for the di-tosylation. As such, eight protons appear between 7.90 and 7.30 ppm corresponding to both tosyl groups. The two corresponding methyl protons presented as a singlet at 2.45 ppm. Compound **22** was confirmed by the absence of tosyl NMR signals, and the presence of a triplet at 3.39 ppm, corresponding to both methylene groups bonded to azide groups.

Equimolar cycloaddition of **22** with previously prepared propargyl coumarin **4** afforded **23** with very good yields. Proton spectrum presented in the aromatic region the protons belonging to the coumarin (between 7.70 and 6.20 ppm) and the triazole (8.00 ppm) moieties. The methylene of the coumarin moiety was observed at 5.29 ppm as a singlet, as expected. PEG proton signals were shifted in four different places, where the most deshielded triplet, at 4.60 ppm, corresponded to the methylene bonded to the triazole moiety, and the more shielded one, at 3.39 ppm corresponded to the methylene bonded to the azide moiety. Carbon spectrum sustained these results. FT-IR spectrum showed the azide band at 2106 cm^−1^, indicating that unwanted di-cycloaddition was not obtained.

Subsequent cycloaddition on the other side of the polymer, either using propargylated galactopyranose **10** or propargylated mannopyranose **11**, led to the formation of glycoconjugates **24** or **25**, respectively, in 11% and 17% yields. Structure confirmation by proton NMR was difficult due to the overlap of PEG and carbohydrate signals, while MALDI results were in accordance with the expected estimated values (estimated medium value of 1426 *m*/*z*, obtained 1511 *m*/*z* for compound **24**, and 1477 *m*/*z* for compound **25**).

Diesterification of PEG with coumarin alkyne **3** using DCC afforded a new fluorescent polymer, **26** (Scheme 9). Product **26** was easily identified by proton NMR, where the relative intensity of coumarin signals, compared to PEG methylene signals, were as expected for the diester. As such, 4 aromatic protons at 7.47 and 6.98 ppm corresponded to both coumarins. PEG proton signals were identified at 4.23 and 3.66 ppm, with a relative intensity for 86 protons. Carbon spectrum analysis was also consistent with the proposed structure (see experimental part).

Subsequent double azide-alkyne Huisgen cycloaddition with thymidine azide **16** using a catalytic amount of CuI and DIPEA in THF, afforded the symmetric polymer **27**. Structural characterization was observed to be in agreement with the proposed structure. More precisely, proton NMR spectrum showed new triazole protons at 7.77 ppm and coumarin signals in the expected aromatic region at 7.41 and 6.98 ppm. Thymidine signals were observed at 7.30, 6.24, and 4.47 ppm, and PEG signals at 4.20 and 3.60 ppm. All the relative intensity for the assigned protons were in agreement with bis-thymidine cycloaddition. The carbon spectrum confirmed these results, mainly by the presence of a triazole signal at 135.5 ppm, the anomeric carbon at 84.9 ppm and the PEG main carbon chain at 70.5 ppm. The MALDI-TOF expected medium value was 2132 *m*/*z*, and it was obtained 2153 *m*/*z* for compound **27**, with sizes ranging from 1932 to 2461 *m*/*z*, in which the increment observed was due to ethylene glycol monomer.

### 2.5. PLGA Derivatives

PLGA is a hydrophobic, linear and biocompatible polymer, approved by the FDA and EMA for biological applications [43]. Upon hydrolysis, PLGA is able to release pharmaceutical agents, and the resulting lactic and glycolic acids are metabolized on the Krebs cycle [44]. A polymer with a molecular weight range between 7000 and 17,000 g/mol was chosen with a 50:50 glycolic/lactic acid ratio, which translates into a more amorphous final structure.

PLGA glycoconjugate **28** was prepared by amide formation between **14** and PLGA using methanesulfonic acid (Scheme 10). Due to the fact that amines are more reactive than alcohols, it was assumed that an amide was the major product of this reaction, although esterification may have also occurred.

Coumarin **8** reacted with PLGA in the presence of methanesulfonic acid (Scheme 11) to afford the fluorescent amide, **29**. 

Proton NMR analysis of compound **28** identified the methyl, methylene, and methine groups that form the polymer at 1.58, 4.83, and 5.22 ppm, respectively. The characteristic signal from the triazole moiety can be identified at 7.01 ppm, and a few sugar signals between 4.43 and 4.11 ppm. For polymer **29** it was also possible to identify the methyl, methylene, and methine groups from the polymer chain at 1.58, 4.83, and 5.24 ppm, respectively, plus the coumarin and triazole characteristic protons at 8.29, 7.80, and 7.54 ppm. Identification of these ligands was very difficult, due to high NMR signal accumulation from the polymer structure. Nevertheless, it was possible to identify very distinctive signals for both polymers. MALDI-TOF analysis could not be made for these polymers due to their very high polydispersity. Correlation between zeta potentials of their respective nanoparticles, and these observed NMR signals allowed us to conclude that the functionalization had occurred. 

### 2.6. Nanoparticles Preparation and Characterization

Polymeric glycoconjugates were transformed into nanoparticles with the glycosides towards their exterior. Oil in water emulsification/solvent evaporation technique is an easy and highly reproducible method for nanoparticle preparation, and it was our choice for the transformation of the prepared polymers into nanoparticles.

Nanoparticles from glycoconjugate **20** were prepared and analyzed by Scanning Electron Microscopy (SEM) and the results are presented in Figure 1A,B. This glycoconjugate formed mostly a polymeric film upon deposition, with some dispersed agglomerates. The size of the agglomerates ranged between 220 and 580 nm.

Glycoconjugates **24** and **25** were also transformed into nanoparticles and analyzed by SEM, and are presented in Figure 1C–F. Galactoconjugate **24** formed more agglomerates than glycoconjugate **20**, and a film was still present. For the mannoconjugate **25,** it was observed that it formed irregular films, with multiple cracks, and a high degree of convolution of neighboring agglomerates either by the polymer’s preferential formation of films, or due to film deposition on top of those agglomerates. Dynamic Light Scattering (DLS) technique (results presented in Figure 2), where the particles are suspended in water in a colloidal manner, showed average particle size of 253 (**20**), 118 (**24**), and 193 nm (**25**), peak size of 219 (**20**), 105 (**24**), and 171 nm (**25**) and high polydispersity indexes of 5.828 (**20**), 0.440 (**24**), and 0.682 (**25**). These results confirmed that PEG derivatives formed polydisperse samples, with aggregates. 

PLGA glucoconjugate **28** was capable of forming spherical nanoparticles, with a smooth surface and with practically no aggregation as seen in Figure 3A,B. The particles presented a size range of 114–234 nm. DLS gave similar results, with an average particle size of 189 nm, peak size 171 nm (Figure 4A) and a low polydispersity index of 0.174. The zeta potential was also measured, with a value corresponding to −28.2 mV.

Nanoparticles from the coumarin-containing PLGA derivative, **29**, presented a spherical and smooth surface, with no aggregation (Figure 3C,D) and size range of 174–289 nm. DLS results showed an average particle size of 273 nm, peak size 219 nm (Figure 4B), a polydispersity value of 0.146, and a zeta potential of −56.0 mV. This high zeta potential value may be explained by the presence of both the triazole ring and amide bonds towards the exterior of the particles, as expected by the technique employed for their formation.

## 3. Materials and Methods

All reagents used were purchased from Sigma-Aldrich/Merck, Carbosynth or Fluka. The solvents used as reaction media were dried according to the procedures described in the literature [45]. Briefly, DMF was dried with barium oxide, filtered, distilled and stored with 3Å molecular sieves. Dichloromethane (DCM) was distilled from calcium hydride under an argon atmosphere and used immediately. Propargyl alcohol was dried overnight with potassium carbonate, distilled and used immediately. PEG was purchased from Sigma-Aldrich, with an average molecular weight of 1000. D,L-poly(lactic-glycolic acid) (PLGA) 50:50, with a molecular weight range of 7000–17,000 was purchased from Sigma-Aldrich. Column chromatography was performed using Carlo Erba’s silica gel, 40–63 µm mesh. Preparative TLC was performed using silica gel 60GF DGF254 purchased from Macherey-Nagel.

### 3.1. Methods for Compounds Characterization

Melting points were measured using an Electrothermal Melting Point Apparatus. Optical rotations were measured using a Bellingham and Stanley Ltd (Department of Chemistry FCT-NOVA; Portugal). ADP410 Polarimeter, at sodium D line (Department of Chemistry FCT-NOVA; Portugal). Nuclear magnetic resonance spectra were obtained using a Bruker ARX 400 MHz (Department of Chemistry FCT-NOVA; Portugal) for ^1^H and 101 MHz for ^13^C. FT-IR spectra were recorded using a Bruker Tensor 27 (Department of Chemistry FCT-NOVA; Portugal). UV spectra were recorded on a Perkin Elmer Lambda 35 (Department of Chemistry FCT-NOVA; Portugal, using a quartz cell (Department of Chemistry FCT-NOVA; Portugal). Emission data were recorded on a Perkin Elmer LS 45 Luminescence Spectrometer, using a quartz cell (Department of Chemistry FCT-NOVA; Portugal). High resolution mass spectra were recorded on a Bruker Microtof using electrospray ionization, flow injection analysis and time-of-flight detector. MALDI spectra were recorded on a Bruker Ultraflex III TOF/TOF using HCCA as the matrix (Unidade de Espectrometria de Massas e Proteómica; Spain). DLS and zeta potential measurements were performed using Horiba Scientific Nano Particle Analyzer SZ-100 (Department of Chemistry FCT-NOVA; Portugal) with 1 mg of particles in 1 mL of filtered, ultra-pure water, at 25 °C. SEM micrographs were obtained with a JEOL JSM 7001 scanning electron microscope (Electron Microscopy Laboratory of Instituto Superior Técnico; Portugal) with an accelerating voltage of 10 kV, after chromium-coating under argon atmosphere the previously fixed samples onto metallic studs.

### 3.2. Nanoparticles Preparation

Polymer matrix (50 mg) was dissolved in 5 mL of DCM and poured into 8.0 mL of aqueous PVA 2%. The resulting oil-in-water preparation was sonicated at 60 W for a minute, in periods of 10 s. The resulting emulsion was magnetically stirred overnight. The resulting suspension was centrifuged, and the pellet was washed three times with deionized water.

### 3.3. Synthetic Procedures

#### 3.3.1. Ethyl 3-(7-Hydroxy-4,8-Dimethyl-2-oxo-2H-Chromen-3-yl)propanoate (**1**)

3-Methylresorcinol (11.020 g, 96.9 mmol) was dissolved in dry ethanol (50 mL) with magnetic stirring. Diethyl 2-acetylglutarate (21.0 mL, 96.7 mmol) was added and the reaction flask was cooled to 0 °C and dried hydrogen chloride was passed through the flask for 3 h, and the reaction mixture was maintained at ambient temperature for 16 h. Cool water (500 mL) was added and the resulting precipitate was collected and dried, affording **1** (16.833 g, 60%) as a beige solid. m.p. = 145–148 °C; FT-IR (NaCl) ῡ_max_: 3263 (br, OH st), 2983 (s, C-H w), 1731–1703 (s, O-C=O st), 1619 (br, C=C m) cm^−^^1^; ^1^H NMR (400 MHz, CDCl_3_) δ: 7.27 (d, *J* = 8.7 Hz, 1H, H-5), 6.78 (d, *J* = 8.7 Hz, 1H, H-6), 4.15 (q, *J* = 7.1 Hz, 2H, H-16), 2.98 (t, *J* = 7.6 Hz, 2H, H-13), 2.63 (t, *J* = 7.5 Hz, 2H, H-14), 2.40 (s, 3H, H-12), 2.27 (s, 3H, H-11), 1.26 (t, *J* = 7.2 Hz, 3H, H-17); ^13^C NMR (101 MHz, CDCl_3_) δ: 173.6 (C-15), 162.5 (C-2), 156.9 (C-7), 151.7 (C-9), 148.5 (C-4), 122.5 (C-5), 120.2 (C-3), 113.8 (C-10), 111.9 (C-6), 111.5 (C-8), 60.8 (C-16), 32.9 (C-14), 23.2 (C-13), 15.0 (C-12), 14.2 (C-17), 8.0 (C-11).

#### 3.3.2. Ethyl 3-(4,8-Dimethyl-2-oxo-7-(Prop-2-yn-1-yloxy)-2H-Chromen-3-yl)propanoate (**2**)

Compound **1** (0.992 g, 3.4 mmol) was dissolved in dry acetone (8.3 mL), and anhydrous potassium carbonate (1.665 g, 12.1 mmol) was added. The reaction mixture was kept under constant stirring for 30 min, and then propargyl bromide (0.5 mL, 4 mmol) was added and the mixture was refluxed for 4 h. After cooling to room temperature, the solvent was evaporated and the crude product was dissolved in DCM (10 mL), washed four times with deionized water (4 × 15 mL), dried over anhydrous sodium sulfate, filtered and evaporated, affording **2** (1.093 g, 98%) as a beige solid m.p. = 151–152 °C; ^1^H NMR (400 MHz, CDCl_3_) δ: 7.45 (d, *J* = 8.9 Hz, 1H, H-5), 6.96 (d, *J* = 8.9 Hz, 1H, H-6), 4.80 (d, *J* = 2.4 Hz, 2H, H-1’), 4.12 (q, *J* = 7.1 Hz, 2H, H-16), 2.97 (t, *J* = 7.6 Hz, 2H, H-13), 2.60 (t, *J* = 7.7 Hz, 2H, H-14), 2.54 (t, *J* = 2.5 Hz, 1H, H-3’), 2.43 (s, 3H, H-12), 2.32 (s, 3H, H-11), 1.24 (t, *J* = 7.1 Hz, 3H, H-17); ^13^C NMR (101 MHz, CDCl_3_) δ: 172.9 (C-15), 161.7(C-2), 157.4 (C-7), 151.4 (C-9), 147.4 (C-4), 122.4 (C-5), 121.8 (C-3), 115.0 (C-10), 114.7 (C-8), 108.2 (C-6), 78.2 (C-2’), 76.0 (C-3’), 60.5 (C-16), 56.4 (C-1’), 32.7 (C-13), 23.2 (C-14), 15.0 (C-12), 14.21 (C-17), 8.3 (C-11).

#### 3.3.3. 3-(4,8-Dimethyl-2-oxo-7-(Prop-2-yn-1-yloxy)-2H-Chromen-3-yl)propanoic Acid (**3**)

Compound **2** (0.822 g, 2.5 mmol) was dissolved in ethanol (20.0 mL), and 0.25 M sodium hydroxide aqueous solution (12 mL) was added. The reaction was stirred at 80 °C for 1 h, cooled down to room temperature, and deionized water (20 mL) was added. The mixture was acidified with 1 M hydrochloric acid to pH 1 and the precipitate obtained was filtered and the resulting solid was dried, affording **3** (0.712 g, 95%) as a white solid. m.p. = 181–183 °C; FT-IR (NaCl) ῡ_max_:: 3389 (br, OH st), 3055 (s, C-H w), 2943 (s, C-H w), 2228 (br, C≡C st), 1763-1704 (s, O-C=O st), 1641 (br, C=C m) cm^−1^; ^1^H NMR (400 MHz, DMSO-d_6_) δ: 7.63 (d, *J* = 8.9 Hz, 1H, H-5), 7.10 (d, *J* = 9.0 Hz, 1H, H-6), 4.95 (s, 2H, H-1’), 3.61 (t, *J* = 2.7 Hz, 1H, H-3’), 2.77 (t, *J* = 7.8 Hz, 2H, H-13), 2.40–2.35 (m, 4H, H-12,14), 2.18 (s, 3H, H-11); ^13^C NMR (101 MHz, DMSO-d_6_) δ: 173.8 (C-15), 160.7 (C-2), 157.0 (C-7), 150.5 (C-9), 147.7 (C-4), 123.3 (C-5), 121.2 (C-3), 114.3 (C-10), 112.6 (C-8), 108.8 (C-6), 79.0 (C-2’), 78.7 (C-3’), 56.3 (C-1’), 32.6 (C-14), 23.0 (C-13), 14.7 (C-12), 8.0 (C-11).

#### 3.3.4. 7-(Prop-2-yn-1-yloxy)-2H-Chromen-2-One (**4**)

Potassium iodide (2.454 g, 14.7 mmol) and potassium carbonate (2.045 g, 14.7 mmol) were added to a solution of umbelliferone (2.011 g, 12.4 mmol) in DMF (30 mL) under magnetic stirring and argon atmosphere. Then, propargyl bromide 80% in toluene (1.7 mL, 14.8 mmol) was added drop-wise and the reaction was heated at 80 °C. After the completion of the reaction (2 h 30 min), dichloromethane was added (40 mL) and the mixture was transferred to a separating funnel and washed three times with water (3 × 20 mL). The organic phase was dried over anhydrous sodium sulfate, filtered and concentrated to afford 7-(prop-2-yn-1-yloxy)-2*H*-chromen-2-one, **4**, as a light brown solid (2.382 g, 96% without purification) and was used, as such, for further reactions. m.p. = 114–115 °C; UV (DCM) λ_max_ = 319 nm; Em (DCM) λ_max_ = 386 (λ_exc_ = 320 nm); FT-IR (NaCl) ῡ_max_: 3266 (C-H st), 3084 (ar C-H st), 2119 (C=C st), 1701 (C=O st), 1614 (C=C st) cm^−1^; ^1^H NMR (400 MHz, CDCl_3_) δ: 7.65 (d, *J*=9.5 Hz, 1H, H-4), 7.41 (d, *J*=8.5 Hz, 1H, H-5), 7.02–6.81 (m, 2H, H-6 and H-8), 6.28 (d, *J* = 9.5 Hz, 1H, H-3), 4.77 (d, *J* = 2.3 Hz, 2H, H-9), 2.58 (t, *J* = 2.2 Hz, 1H, H-11); ^13^C NMR (101 MHz, CDCl_3_) δ: 161.0 (C-2), 160.6 (C-7), 155.7 (C-8a), 143.3 (C-4), 128.8 (C-5), 113.7 (C-3), 113.2 (C-4a), 113.1 (C-6), 102.2 (C-8), 77.4 (C-10), 76.6 (C-11), 56.2 (C-9).

#### 3.3.5. Prop-2-yn-1-yl 2-oxo-2H-Chromene-3-Carboxylate (**5**)

DCC (2.600, 12.6 mmol) and DMAP (0.129 g, 1.1 mmol) were added to a solution of coumarin-3-carboxylic acid (2.000 g, 10.5 mmol) in DCM (30 mL). The reaction mixture was stirred magnetically under an argon atmosphere and propargyl alcohol (1.2 mL, 21.0 mmol) was added drop-wise. After stirring overnight the mixture was filtered, transferred to a separating funnel and washed three times with acetic acid (5%, 3 × 15 mL) and then with water (20 mL). The organic phase was dried over anhydrous sodium sulfate, filtered and the solvent evaporated under vacuum. The residue was washed with petroleum ether to afford the desired product as a white solid (2.399 g quantitative yield without purification) and used for further reactions. m.p. = 125-127 °C; UV (DCM) λ_max_ = 294 nm; Em (DCM) λ_max_ = 418 (λ_exc_ = 320 nm); FT-IR (NaCl) ῡ_max_:: 3315, 3228 (C≡C-H st), 3058 (ar C-H st), 2936, 2852, 2117 (C≡C st), 1717 (C=O), 1612 (C=C st) cm^−1^; ^1^H NMR (400 MHz, CDCl_3_) δ: 8.61 (s, 1H, H-4), 7.87–7.45 (m, 2H, H-5 and H-7), 7.36 (m, 2H, H-6 and H-8), 4.95 (d, *J* = 2.1 Hz, 2H, H-9), 2.56 (t, *J* = 2.0 Hz, 1H, H-11); ^13^C NMR (101 MHz, CDCl_3_) δ: 162.1 (C-9), 156.4 (C-2), 155.3 (C-8a), 149.6 (C-7), 134.8 (C-5), 129.7 (C-4), 125.0 (C-6), 117.7 (C-4a), 117.2 (C-8), 116.9 (C-3), 75.7 (C-12), 53.2 (C-10). C-11 is probably obscured by the CDCl_3_ signal.

#### 3.3.6. 4-Methyl-7-(Prop-2-yn-1-yloxy)-2H-Chromen-2-One (**6**)

7-Hydroxy-4-methyl coumarin (3.532 g, 20.1 mmol) was dissolved in dry acetone (30 mL) and anhydrous potassium carbonate (3.209 g, 20.2 mmol) was added. The reaction was magnetically stirred for 30 min, and propargyl bromide (2.7 mL, 24.1 mmol) was added. The reaction flask was heated at 50 °C and reacted for 18 h. The solvent was evaporated, and the residue was dissolved in DCM (20 mL) and washed four times with water (20 mL). The organic phase was dried over anhydrous sodium sulfate, filtered and evaporated, to afford product **6** (3.786 g, 88%) as a beige solid. m.p. = 132–134 °C; FT-IR (NaCl) ῡ_max_: 3304 (s, ≡C-H, st), 3069 (s, C-H m), 2924 (s, C-H m), 1764–1721 (s, O-C=O st), 1619 (br, C=C m), 1262 (s, CO-O m) cm^−1^; ^1^H NMR (400 MHz, CDCl_3_) δ: 7.52 (d, *J* = 9.5 Hz, 1H, H-5), 7.00–6.80 (m, 2H, H-6,8), 6.15 (s, 1H, H-3), 4.76 (d, *J* = 2.5 Hz, 2H, H-1’), 2.59 (t, *J* = 2.5 Hz, 1H, H3’), 2.40 (s, 3H, H-11); ^13^C NMR (101 MHz, CDCl_3_) δ: 161.1 (C-2), 160.4 (C-7), 155.0 (C-4), 152.5 (C-9), 125.7 (C-5), 114.3 (C-10), 112.7 (C-6), 112.4 (C-3), 102.2 (C-8), 77.5 (C-2’), 76.5 (C-3’), 56.2 (C-1’), 18.7 (C-11).

#### 3.3.7. 2-Azidoethanamine (**7**)

2-Bromoethylamine hydrobromide (6.162 g, 30.1 mmol) was added to an aqueous sodium azide solution (3.5 M, 26 mL). The reaction was kept at 85 °C for 21 h, and then cooled to 0 °C. Potassium hydroxide (7.985 g, 142.3 mmol) was then added in three portions and the pH reached 10. The compound was extracted with diethyl ether (4 × 25 mL) and the organic phase was dried over anhydrous sodium sulfate, filtered and the solvent evaporated at 28 °C to a green mixture of 2.688 g containing 2.105 g (24.4 mmol, 81%) of the desired compound in ether. The yield of this reaction was determined by NMR. The solvent was never fully evaporated due to the sensitive and explosive nature of **7**. FT-IR (NaCl) ῡ_max_: 3447 (br, N-H st), 2939 (s, C-H w), 2093 (s, N_3_ st) cm^−1^; ^1^H NMR (400 MHz, CDCl_3_) δ: 3.36 (t, *J* = 5.7 Hz, 2H, H-2), 2.87 (t, *J* = 5.7 Hz, 2H, H-1), 1.71 (s, 2H, NH_2_); ^13^C NMR (101 MHz, CDCl_3_) δ: 54.7 (C-2), 41.4 (C-1).

#### 3.3.8. 7-((1-(2-Aminoethyl)-1H-1,2,3-Triazol-4-yl)methoxy)-4-Methyl-2H-Chromen-2-One (**8**)

Compound **7** (0.534 g, 6.2 mmol) was dissolved in freshly distilled THF (6.0 mL). DIPEA (0.2 mL), copper iodide (0.044 g, cat.) and **6** (2.657 g; 12.4 mmol) were added. The mixture was kept under magnetic stirring for 1.5 h. The mixture was purified by silica gel chromatography using chloroform/methanol (8: 2) as eluent, affording **8** as a yellow solid (1.396 g, 75%). m.p. = 114–116 °C; ^1^H NMR (400 MHz, DMSO-d_6_) δ: 8.26 (s, 1H, H-5), 7.69 (d, *J* = 8.8 Hz, 1H, H-5’), 7.15 (d, *J* = 2.5 Hz, 1H, H-3’), 7.04 (dd, *J* = 8.8, 2.4 Hz, 1H, H-6’), 6.22 (t, *J* = 1.3 Hz, 1H, H-8’), 5.26 (s, 2H, H-6), 4.34 (t, *J* = 6.2 Hz, 2H, H-2), 2.97 (t, *J* = 6.2 Hz, 2H, H-1), 2.40 (d, *J* = 1.3 Hz, 3H, H-11’), 1.90 (s, 2H, NH_2_); ^13^C NMR (101 MHz, DMSO-d_6_) δ: 161.1 (C-2’), 160.1 (C-7’), 154.7 (C-4’), 153.4 (C-9’), 141.7 (C-4), 126.5 (C-5’), 125.1 (C-5), 113.4 (C-14’), 112.6 (C-6’), 111.3 (C-8’), 101.6 (C-3’), 61.7 (C-6), 54.9 (C-2), 52.5 (C-1), 18.1 (C-11’); MS (ESI+) *m*/*z*: calc: for C_15_H_17_N_4_O_3_ (M + H^+^) 301.1295, found 301.1299.

#### 3.3.9. 1-O-Propargyl-D-Glucopyranoside (**9**)

D-glucose (4.149 g, 23.0 mmol) was dissolved in propargyl alcohol (7.8 mL, 132.0 mmol), and sulfuric acid immobilized in silica gel (150.4 mg) was added. The reaction mixture was heated at 65 °C overnight. After cool down to room temperature, the mixture was purified by silica column chromatography using an ethyl acetate/acetone/water (10:10:1) mixture, affording **9** as a mixture of α and β anomers, 2:1 ratio. Sulfuric acid supported on silica was prepared accordingly with the procedure described by Roy and Mukhopadhyay [36]. For detailed characterization, the α, β mixture **9** was per-*O*-acetylated using acetic anhydride to give the α, β-propargylated per-acetylated glucopyranose **12.**

#### 3.3.10. 1-O-Propargyl-D-Galactopyranoside (**10**)

D-galactose (2.000 g, 11.1 mmol), propargyl alcohol (3.2 mL, 55.6 mmol) and sulfuric acid supported on silica (72.5 mg) were mixed together under magnetic stirring and an argon atmosphere. The reaction flask was heated at 65 °C overnight, and after cooling the mixture was chromatographed [ethyl acetate (AcOEt), then 10:10:1 AcOEt/acetone/water] affording 45% of propargylated galactopyranose **10** as a yellowish oil (1.099 g, 0.5 mmol, 2:1 α/β ratio). Sulfuric acid supported on silica was prepared accordingly with the procedure described by Roy and Mukhopadhyay [36]. ${\left[\alpha \right]}_{D}^{16}=$ +110.0° (C=1.4, MeOH); FT-IR (NaCl), ῡ_max_: 3382 (br, O-H st), 2925 (w, C-H st) cm^−1^; ^1^H NMR (400 MHz, D_2_O) δ: 5.04 (s, 2H, H-1α), 4.50 (d, *J* = 7.9 Hz, 1H, H-1β), 4.43–4.39 (m, 1H), 4.26 (dd, *J* = 5.8, 2.2 Hz, 4H), 3.92 (s, 2H), 3.91–3.83 (m, 3H), 3.81–3.74 (m, 5H), 3.72–3.63 (m, 7H), 3.59 (dd, *J* = 9.9, 3.3 Hz, 1H), 3.50–3.41 (m, 1H), 2.86–2.80 (m, 1H); ^13^C NMR (101 MHz, D_2_O) δ: 101.1, 97.3, 79.0, 78.9, 76.2, 75.9, 75.3, 72.7, 71.2, 70.5, 69.4, 69.1, 68.6, 67.9, 60.9, 60.9, 56.5, 54.8.

#### 3.3.11. 1-O-Propargyl-D-Mannopyranoside (**11**)

D-mannose (2.000 g, 11.1 mmol), propargyl alcohol (3.2 mL, 55.6 mmol) and sulfuric acid supported on sílica (58 mg) were mixed together under magnetic stirring and argon atmosphere. The reaction flask was heated at 65 °C overnight, and after cooling the mixture was purified by column chromatography (AcOEt, then 10:10:1 AcOEt/acetone/water) affording propargylated mannopyranose **11** as a yellowish oil (0.928 g, 38%, 2:1 α/β ratio). Sulfuric acid supported on silica was prepared accordingly to the procedure described by Roy and Mukhopadhyay [36]. FT-IR (NaCl), ῡ_max_: 3289 (br), 2930 (w) cm^−1^; ^1^H NMR (400 MHz, D_2_O) δ: 5.00 (s, 1H, H-1α), 4.40–4.20 (m, 2H, H-1’), 3.92 (s, 1H), 3.89–3.81 (m, 1H), 3.81–3.69 (m, 2H), 3.69–3.57 (m, 2H), 2.89 (s, 1H, C-3’); ^13^C NMR (101 MHz, D_2_O) δ: 98.7 (C-1α), 79.0, 76.3, 73.1, 70.5, 69.9, 66.6, 60.8, 54.6 (C-1’).

#### 3.3.12. 1-O-Propargyl-2,3,4,6-Tetra-O-Acetyl-D-Glucopyranoside (**12**)

Glucopyranoside **9** (3.418 g, 15.7 mmol) reacted with acetic anhydride (16.2 mL, 171.6 mmol) in dry pyridine (30.0 mL) for 4 h under an argon atmosphere. The solvent was then evaporated and the remaining solid was dissolved in dichloromethane and purified by silica column chromatography using hexane/ethyl acetate (1:1) affording **12** as a colorless oil (4.998 g, 82%). FT-IR (NaCl) ῡ_max_: 3278 (s, ≡C-H w), 2961 (s, C-H w), 1749 (s, O-C=O st) cm^−1^; ^1^H NMR (400 MHz, CDCl_3_) δ: 5.49 (t, *J* = 9.8 Hz, 1.17H, H-3), 5.30–5.27 (m, 1H, H-1α), 5.15–4.97 (m, 2H), 4.92 (dd, *J* = 10.3, 3.8 Hz, 1H), 4.78 (d, *J* = 8.0 Hz, 0.42H H-1β), 4.38 (d, *J* = 2.3 Hz, 1H), 4.28 (d, *J* = 2.4 Hz, 2H, H-1’), 4.26 (d, *J* = 4.1 Hz, 1H), 4.18–4.03 (m, 3H), 3.78–3.70 (m, 0.42H, H-5β), 2.48 (t, *J* = 2.3 Hz, 0.41H, H-3’β), 2.46 (t, *J* = 2.3 Hz, 1H, H-3’α), 2.06 (s, 19H, CH_3_).

#### 3.3.13. 1-[1’-Ethylamine-Triazolyl-4]-1-Methyl-2,3,4,6-Tetra-O-Acetyl-D-Glucopyranoside (**13**)

Compound **7** (0.037 g, 0.4 mmol) was dissolved in 4,0 mL of freshly distilled THF, and DIPEA (0.1 mL, 0.6 mmol), copper iodide (0.027 g, cat.) and **12** (0.237 g, 0.6 mmol) were added. After 2.5 h the solvent was evaporated and the mixture was purified by column chromatography using chloroform/methanol (9:1), affording **13** (0.138, 73%) as a yellow solid. m.p. = 60–65 °C (degradation); FT-IR (NaCl) ῡ_max:_ 3421 (br, NH st), 1745 (s, O-C=O st) cm^−^^1^; ^1^H NMR (400 MHz, CDCl_3_) δ: 7.68 (s, 1H, H_α_-5’), 7.64 (s, 0.48H, H_β_-5’), 5.49 (t, *J* = 9.8 Hz, 1H, H-3), 5.22 (d, *J* = 4.1 Hz, 1H, H-1_α_), 5.09 (s, 1H, H-2), 4.91–4.83 (m, 3H,), 4.70 (d, *J* = 12.5 Hz, 2H, H-1_β_,6), 4.44 (t, *J* = 5.9 Hz, 3H, H-7’), 4.31–4.22 (m, 2H), 4.11 (dd, *J* = 10.1, 3.1 Hz, 2.38H), 3.75 (d, *J* = 7.8 Hz, 0.8H, H-5), 3.26 (s, 3H, H-8’), 2.13–1.98 (m, 20H, CH_3_); ^13^C NMR (101 MHz, CDCl_3_) δ: 170.7–169.5 (C-Ac), 144.1 (C-4’β), 143.5 (C-4’α), 123.7 (C-5’β), 123.6 (C-5’α), 99.9 (C-1α), 95.0 (C-1β), 72.8–72.0 (C-5), 71.9–71.3 (C-2), 70.7–70.6 (C-3), 68.5–68.3 (C-6), 67.5 (C-4), 63.0 (C-6’), 61.8 (C-6β), 61.3 (C-6α), 60.8 (C-7’), 41.7 (C-8’), 20.8–20.6 (CH_3_).

#### 3.3.14. 1-[1’-Ethylamine-Triazolyl-4]-1-O-Methyl-D-Glucopyranoside (**14)**

Compound **13** (0.110 g, 0.2 mmol) was dissolved in dry methanol at ice-bath temperature, and sodium methoxide was added (0.012 g, 0.2 mmol). The reaction was allowed to warm to room temperature and stirred for a further 3 h. Pre-activated and thoroughly washed acidic Dowex was added until the pH changed from basic to acidic. The solution was filtered and the solvent evaporated to give compound **14** (0.071 g, 0.2 mmol, 100%) as a white foam. Due to the complex anomeric mixture this compound’s characterization was based on the spectra of per-*O*-acetyl compound **13**.

#### 3.3.15. 5’-O-[(4-Methylphenyl)sulfonyl] Thymidine (**15**)

In a round bottom flask 2’-deoxythymidine (1.569 g, 6.5 mmol) was dissolved in dry pyridine (7.5 mL) and the flask cooled in an ice bath for 1 h. Tosyl chloride (1.236 g, 6.5 mmol) dissolved in dry pyridine (2.5 mL) was added dropwise and the reaction allowed to attain room temperature during 17 h. The solvent was then evaporated, and the resulting crude product was dissolved in ethanol at 75 °C and then cooled down to −10 °C for 12 h. The resulting crystals of **15** were filtered and washed with cold ethanol (3 × 1.0 mL, 1.889 g, 54%). m.p. = 189–191 °C; FT-IR (NaCl) ῡ_max_: 3374 (br, OH st), 3067 (s, C-H w), 2924 (s, C-H w), 1724–1694 (s, N-C=O st) cm^−^^1^; ^1^H NMR (400 MHz, DMSO-d_6_) δ: 11.31 (s, 1H, NH), 7.79 (d, *J* = 8.2 Hz, 2H, H-12), 7.47 (d, *J* = 8.1 Hz, 2H, H-13), 7.38 (s, 1H, H-6), 6.15 (t, *J* = 6.9 Hz, 1H, H-1), 5.43 (d, *J* = 4.2 Hz, 1H, OH), 4.26 (dd, *J* = 10.9, 3.2 Hz, 1H, H-3), 4.19–4.15 (m, 2H, H-5), 3.86 (dt, *J* = 6.8, 3.5 Hz, 1H, H-4), 2.41 (s, 3H, H-15), 2.19–2.04 (M, 2H, H-2), 1.77 (s, 3H, H-8); ^13^C NMR (101 MHz, DMSO-d_6_) δ: 163.6 (C-9), 150.4 (C-10), 145.1 (C-11), 135.9 (C-14), 132.1 (C-6), 130.2 (C-13), 127.6 (C-12), 109.8 (C-7), 84.0 (C-1), 83.2 (C-4), 70.1 (C-3), 69.9 (C-5), 38.3 (C-2), 21.1 (C-15), 12.1 (C-8).

#### 3.3.16. 5’-Azide-5’-Deoxythymidine (**16**)

Method I—conventional heating: compound **15** (0.562 g, 1.4 mmol) was dissolved in DMF (5.0 mL), and sodium azide was added (0.407 g, 6.4 mmol). The reaction mixture was heated to 85 °C on an oil bath for 15 h. The solvent was then evaporated under reduced pressure and the mixture was purified by silica gel column chromatography using chloroform/methanol (9:1) as eluent, affording **16** (0.325 g, 87%) as a beige solid.

Method II—microwave irradiation: compound **15** (0.817 g, 2.1 mmol) was dissolved in DMF (5.0 mL), and sodium azide (0.620 g, 10.3 mmol) was added. The mixture was irradiated with microwaves with a power of 250 W at a temperature of 100 °C, for two 1 min cycles. The solvent was evaporated under reduced pressure and the mixture was purified by column chromatography (chloroform/methanol, 9:1) to afford **16** (0.505 g, 90%) as a beige solid. m.p. = 156–158 °C; FT-IR (NaCl) ῡ_max_: 3422 (br,OH st), 2926 (s, C-H w), 2103 (s, N_3_ st), 1701–1685 (s, N-C=O st) cm^−1^; ^1^H NMR (400 MHz, CD_3_OD) δ: 7.56 (s, 1H, H-6), 6.29 (t, *J* = 6.8 Hz, 1H, H-1), 4.38 (dt, *J* = 6.5, 4.1 Hz, 1H, H-3), 3.99 (q, *J* = 4.0 Hz, 1H, H-4), 3.63 (qd, *J* = 13.2, 4.4 Hz, 2H, H-5), 2.37–2.25 (m, *J* = 6.9 Hz, 2H, H-2), 1.92 (s, 3H, H-8); ^13^C NMR (101 MHz, CD_3_OD) δ: 166.3 (C-9), 152.3 (C-10), 137.8 (C-6), 111.9 (C-7), 86.4 (C-4), 86.3 (C-1), 72.5 (C-3), 53.4 (C-5), 40.2 (C-2), 12.5 (C-8).

#### 3.3.17. α-Coumarin 3-Carboxylate-ω-Hydroxyl PEG_1000_ (**17**)

Coumarin 3-carboxylic acid (0.456 g, 2.4 mmol) was dissolved in DCM (15 mL) with magnetic stirring and under an argon atmosphere. DCC (0.990 g, 4.8 mmol) and DMAP (0.293 g, 2.4 mmol) were added. After 45 min, a white precipitate was observed and PEG_1000_ (2.000, 2.0 mmol) was added. The reaction flask was heated at 40 °C for 18 h. Then, the mixture was filtrated and washed two times with aqueous acetic acid 5% (2 × 20 mL) and once with water (20 mL). The organic phase was dried over sodium sulfate, filtered, concentrated and purified by column chromatography (chloroform to 9:1 chloroform/methanol gradient), affording the required product in 98% yield (2.307 g, 1.97 mmol) as a yellowish wax. FT-IR (NaCl), ῡ_max_: 3512 (br, O-H st), 2870 (s, C-H st), 1766 (s, C=O st) cm^−1^; UV (DCM) λ_max_ = 292 nm; Em (DCM) λ_max_ = 386 nm (λexc = 320 nm); ^1^H NMR (400 MHz, CDCl_3_) δ: 8.57 (s, 1H, H-4), 7.65 (m, 2H, H-Ar), 7.36 (m, 2H, H-Ar), 4.50 (t, *J* = 4.6 Hz, 2H, O-CH_2_-PEG), 3.85 (t, 2H, O-CH_2_CH_2_-PEG), 3.79–3.56 (m, 84H, PEG); ^13^C NMR (101 MHz, CDCl_3_) δ: 162.8 (C-9), 156.6 (C-2), 155.2 (C-8ª), 148.8, 134.4 (C-Ar), 129.6 (C-Ar), 124.9 (C-Ar), 119.9, 119.8 (C-3 and C-4a), 116.8 (C-Ar), 72.5 (PEG), 70.5 (PEG), 70.2 (PEG), 68.9 (O-CH_2_-CH_2_-PEG), 64.8 (O-CH_2_-CH_2_-PEG), 61.6 (PEG).

#### 3.3.18. α-Coumarin 3-Carboxylate-ω-Tosyl PEG_1000_ (**18**)

Polymer **17** (3.968 g, 3.4 mmol) was dissolved in DCM (25 mL) and triethylamine (0.7 mL, 5.1 mmol) with magnetic stirring and argon atmosphere. Tosyl chloride (0.965 g, 5.1 mmol) was then added slowly in portions. After 18 h, the reaction mixture was evaporated and purified by column chromatography (chloroform to 9:1 CHCl_3_/methanol, gradient), to afford **18** as a yellowish wax in 92% yield (4.001 g, 3.0 mmol). FT-IR (NaCl), ῡ_max_: 2872 (s, C-H st), 1761 (s, C=O st), 1108 (s, O=S=O sym st) cm^−1^; UV (DCM) λ_max_ = 285 nm; Em (DCM) λ_max_ = 412.5 nm (λexc = 320 nm); ^1^H NMR (400 MHz, CDCl_3_) δ: 8.57 (s, 1H, H-4), 7.80 (d, *J* = 8.2 Hz, 2H, H-Ar), 7.73–7.57 (m, 2H, H-Ar), 7.35 (m, 4H, H-Ar), 4.50 (t, *J* = 4.8 Hz, 4H, PEG), 4.16 (t, *J* = 4.8 Hz, 2H, PEG), 3.89–3.81 (t, *J* = 4.8 Hz, 4H, PEG), 3.76–3.49 (m, 120H, PEG), 2.45 (s, 3H, CH_3_); ^13^C NMR (101 MHz, CDCl_3_) δ: 162.8 (C-9), 156.6, 155.2, 148.9 (C-4), 144.8, 134.5, 133.0, 129.8, 129.6, 128.0, 124.9, 117.9, 117.8, 116.8, 70.7, 70.6, 70.6, 70.5 (PEG), 69.3 (PEG), 68.9 (PEG), 68.6 (PEG), 64.9 (PEG), 21.6 (C-14).

#### 3.3.19. α-Coumarin 3-Carboxylate-ω-Azide PEG_1000_ (**19**)

Polymer **18** (4.008 g, 3.0 mmol) was dissolved in DMF (32 mL) and sodium azide (0.393 g, 6.0 mmol) was added in portions with stirring under an argon atmosphere. After 18 h, the DMF was evaporated under reduced pressure and water was added to the resulting residue. The mixture was extracted with DCM, dried over anhydrous sodium sulfate, filtered, concentrated and purified by column chromatography (chloroform to 9:1 chloroform/methanol, gradient), to afford the product as an orange wax (3.291 g, 91%), and used for further reactions. FT-IR (NaCl), ῡ_max_: 2871 (s, C-H st), 2104 (w, N_3_ st) 1761 (s, C=O st) cm^−1^; UV (DCM) λ_max_ = 284 nm; Em (DCM) λ_max_ = 412.5 nm (λexc = 320 nm); ^1^H NMR (400 MHz, CDCl_3_) δ: 8.57 (s, 1H, H-4), 7.79–7.56 (m, 2H, H-5 and H-7), 7.46–7.31 (m, 2H, H-6 and H-8), 4.58–4.41 (m, 2H, O-CH_2_-CH_2_-PEG), 4.02–3.33 (m, 90H, PEG); ^13^C NMR (101 MHz, CDCl_3_) δ: 162.8 (C-9), 156.6 (C-2), 155.2 (C-8a), 148.9, 134.5, 129.6, 124.9, 117.9, 117.9, 116.8, 70.6 (PEG), 50.7 (CH_2_N_3_).

#### 3.3.20. α-Coumarin 3-Carboxylate-ω-[(1H-1,2,3-triazol-4-yl)methoxygalactopyranosyl] PEG_1000_ (**20**)

Azide polymer **19** (0.500 g, 0.4 mmol) was dissolved in THF (4.2 mL) then copper iodide (0.039 g, 0.2 mmol), DIPEA (0.14 mL, 0.8 mmol) and compound **10** (0.086 g, 0.4 mmol) were added. The mixture was magnetically stirred under an argon atmosphere for 34 h. After removal of the solvent under reduced pressure, the residue was suspended in DCM and washed twice with water, once with brine, dried over anhydrous sodium sulfate, filtered and concentrated. Purification by preparative chromatography (95:5 chloroform/methanol) afforded **20** as a yellow wax in 18% yield (0.105 g, 0.1 mmol). FT-IR (NaCl), ῡ_max_: 3458 (O-H st), 2876 (C-H st), 1759 (C=O st) cm^−1^. UV (DCM) λ_max_ = 293 nm; Em (DCM) λ_max_ = 413 nm (λexc = 320 nm); ^1^H NMR (400 MHz, CDCl_3_) δ: 8.57 (s, 1H, H-4), 8.23–7.99 (m, 1H, H-9’), 7.72–7.61 (m, 2H, H-5 and H-7), 7.40–7.32 (m, 2H, H-6 and H-8), 5.05 (s, 1H, H-1’), 4.87 (s, 1H, H-6a’), 4.74 (s, 1H (H-6b’), 4.62 (s, 2H, H-3’), 4.50 (s, 2H, H-7’), 4.30–3.34 (m, 100H, PEG); ^13^C NMR (101 MHz, CDCl_3_) δ: 162.8, 156.6, 155.2, 149.2 (C-4), 148.9, 134.5 (C-7), 129.6 (C-5), 124.9 (C-9’ and C-6), 117.9, 117.9, 116.8 (C-8), 98.8 (C-1’), 70.5 (PEG), 69.9 (C-2’), 69.0 (C-4’), 68.9 (PEG), 64.9 (C-7’), 62.1, 60.3 (C-6’), 52.9, 51.1 (C-3’).

#### 3.3.21. α-Tosyl-ω-Tosyl PEG_1000_ (**21**)

PEG (5.000 g, 5 mmol) was dissolved in DCM (50 mL) with magnetic stirring and an argon atmosphere. Triethylamine (1.6 mL, 11.0 mmol) and tosyl chloride (2.383, 12.5 mmol) were added and the reaction mixture was heated at 50 °C for 15 h. The mixture was concentrated under vacuum and purified by column chromatography (chloroform to 9:1 chloroform/methanol gradient) affording quantitatively the product **21** (6.542 g, 100%) as a yellowish wax. FT-IR (NaCl), ῡ_max_: 3511 (br, O-H st), 2873 (s, C-H st) cm^−1^; ^1^H NMR (400 MHz, CDCl_3_) δ: 7.80 (d, *J* = 6.7 Hz, 4H, H-Ar), 7.35 (d, *J* = 6.4 Hz, 4H, H-Ar), 4.16 (s, 4H, 2x CH_2_-OTs), 3.75–3.49 (m, 89H, PEG), 3.38 (s, 2H, PEG), 2.45 (s, 6H, 2x CH_3_); ^13^C NMR (101 MHz, CDCl_3_) δ: 144.8 (C-Ar), 133.1 (C-Ar), 129.8 (C-Ar), 128.0 (C-Ar), 70.8 (PEG), 70.6 (PEG), 70.6 (PEG), 69.3 (PEG), 68.7 (PEG), 21.6 (CH_3_).

#### 3.3.22. α-Azide-ω-Azide PEG_1000_ (**22**)

To a solution of compound **21** (1.488 g, 1.1 mmol) in DMF (18 mL) under magnetic stirring and argon atmosphere was added sodium azide (0.222 g, 3.4 mmol) and the reaction was heated at 50 °C for 16 h. The solvent was evaporated under reduced pressure and then dichloromethane (35 mL) was added and the organic phase was washed twice with brine (2 × 20 mL) and once with water (20 mL), dried over anhydrous sodium sulfate, filtered, concentrated, and purified by column chromatography (chloroform to 9:1 chloroform/methanol stepwise) affording product **22** (0.966 g, 0.9 mmol) as a yellowish wax; ^1^H NMR (400 MHz, CDCl_3_) δ: 3.65 (s, 90H, PEG), 3.39 (t, *J* = 4.5 Hz, 4H, 2× CH_2_-N_3_); ^13^C NMR (101 MHz, CDCl_3_) δ: 70.7 (PEG), 70.7 (PEG), 70.6 (PEG), 70.6 (PEG), 70.0 (PEG), 50.7 (2× CH_2_N_3_).

#### 3.3.23. α-Azide-ω-[7-((1H-1,2,3-Triazol-4-yl)methoxy)-2H-Chromen-2-One)] PEG_1000_ (**23**)

Compounds **22** (0.966 g, 0.9 mmol) and **4** (0.184 g, 0.9 mmol), copper iodide (0.101 g, 0.5 mmol), and DIPEA (0.32 mL, 1.8 mmol) were mixed in THF (8 mL) under magnetic stirring and an argon atmosphere for 18 h. The solvent was evaporated under reduced pressure and then dichloromethane (35 mL) was added. The resulting organic phase was washed three times with water (3 × 20 mL), dried over anhydrous sodium sulfate, filtered, concentrated, and purified by column chromatography (chloroform to 9:1 chloroform/methanol, gradient) affording product **23** (0.947 g, 82%) as an orange wax. FT-IR (NaCl) ῡ_max_: 3553 (w, O-H st) 2872 (s, C-H st), 2106 (w, N_3_ st), 1732 (s, C=O), 1614 (s, C=C bend) cm^−1^; UV (DCM) λ_max_ = 322 nm; Em (DCM) λ_max_ = 386 nm (λexc = 320 nm); ^1^H NMR (400 MHz, CDCl_3_) δ: 8.00 (s, 1H, H-8’), 7.66 (d, *J* = 9.5 Hz, 1H, H-6’), 7.41 (d, *J* = 8.4 Hz, 1H, H-4’), 7.03–6.88 (m, 2H, H-3’ and H-5), 6.26 (d, *J* = 9.5 Hz, 1H, H-5’), 5.29 (s, 2H, H-6), 4.60 (t, *J* = 4.2 Hz, 2H, PEG), 3.90 (t, *J* = 4.6 Hz, 4H, PEG), 3.82–3.54 (m, 82H, PEG), 3.39 (t, *J* = 4.8 Hz, 2H, N_3_-CH_2_); ^13^C NMR (101 MHz, CDCl_3_) δ: 161.4 (C-2’), 161.0 (C-7’), 155.7 (C-8a’), 143.4 (C-4), 142.4 (C-6’), 129.0 (C-4’), 124.8 (C-8’), 113.4 (C-5’), 113.0 (C-4a’), 112.8 (C-5), 102.1 (C-3’), 70.7 (PEG), 70.6 (PEG), 70.5 (PEG), 70.4 (PEG), 70.3 (PEG), 70.0 (PEG), 69.3 (PEG), 62.1 (C-6), 50.7 (2× CH_2_-N_3_).

#### 3.3.24. α–[7-((1*H*-1,2,3-triazol-4-yl)methoxy)-2*H*-Chromen-2-One)]–ω–[(1*H*-1,2,3-triazol-4-yl)methoxygalactopyranosyl] PEG_1000_ (**24**)

Azide **4** (0.150 g, 0.1 mol), propargylated galactopyranosyl **10** (0.039 g, 0.2 mmol), copper iodide (0.011 g, 0.06 mmol), DIPEA (0.04 mL, 0.2 mmol), and 2,2’-bipyridine (0.009 g, 0.06 mmol) were dissolved in THF (1 mL), and mixed together under magnetic stirring and argon atmosphere for 22 h. The mixture was evaporated under reduced pressure, and the residue was dissolved in chloroform (10 mL), washed two times with deionized water (10 mL), once with brine (10 mL), dried over anhydrous sodium sulfate, filtered, concentrated, and purified by column chromatography (chloroform to 9:1 chloroform/methanol, stepwise) to afford **24** (0.019 g, 11%) as an orange wax. λ Abs_max_ = 321 nm; λ Em_max_ = 383.5 nm (λ_exc_ = 320 nm); ^1^H NMR (400 MHz, CDCl_3_) δ: 7.93 (s, 2H, H-11 and H-12), 7.66 (d, *J* = 9.4 Hz, 1H, H-4), 7.40 (d, *J* = 9.3 Hz, 1H, H-5), 7.02–6.91 (m, 2H, H-6 and H-8), 6.27 (d, *J* = 9.4 Hz, 1H, H-3), 5.32–5.24 (m, 2H, H-9), 5.10–5.00 (m, 2H, H-1’), 4.93–4.66 (m, 3H), 4.65–4.47 (m, 4H, H-14), 4.47–4.20 (m, 2H), 4.19–3.99 (m, 3H), 3.95–3.78 (m, 10H, PEG), 3.78–3.39 (m, 120H, PEG), 3.24–3.05 (m, 6H, PEG); ^13^C NMR (101 MHz, CDCl_3_) δ: 161.4 (C-2), 161.1 (C-7), 155.7 (C-8a), 143.4 (C-4), 142.6 (C-10 and C-13), 128.9 (C-5), 124.6 (C-11 and C-12), 113.3 (C-13), 112.9 (C-4a), 112.8 (C-6), 102.1 (C-8), 72.7 (PEG), 70.5 (PEG), 69.3 (PEG), 62.3 (C-9), 61.6 (PEG), 50.4 (C-14).

#### 3.3.25. α–[7-((1*H*-1,2,3-Triazol-4-yl)methoxy)-2*H*-Chromen-2-One)]–ω–[(1*H*-1,2,3-triazol-4-yl)methoxymannopyranosyl] PEG_1000_ (**25**)

Azide **4** (0.153 g, 0.1 mol), propargylated mannopyranosyl **11** (0.055 g, 0.3 mmol), copper iodide (0.033 g, 0.18 mmol), DIPEA (0.04 mL, 0.2 mmol) and 2,2’-bipyridine (0.013 g, 0.08 mmol) were dissolved in THF (1 mL), and mixed together under magnetic stirring and argon atmosphere for 22 h. The mixture was evaporated under reduced pressure, and the residue was dissolved in chloroform (10 mL), washed two times with deionized water (10 mL), once with brine (10 mL), dried over anhydrous sodium sulfate, filtered, concentrated, and purified by column chromatography (chloroform to 9:1 chloroform/methanol, stepwise) to afford **25** (0.030 g, 17%) as an orange wax. λ Abs_max_ = 321 nm; λ Em_max_ = 386.5 nm (λ_exc_ = 320 nm); ^1^H NMR (400 MHz, CDCl_3_) δ: 8.01–7.81 (m, 2H, H-11 and H-12), 7.66 (d, *J* = 9.5 Hz, 1H, H-4), 7.41 (d, *J* = 8.8 Hz, 1H, H-5), 7.05–6.88 (m, 2H, H-6 and H-8), 6.27 (d, *J* = 9.4 Hz, 1H, H-3), 5.35–5.22 (m, 2H, H-9), 5.11–4.90 (m, 2H, H-1’), 4.85–4.64 (m, 3H), 4.63–4.48 (m, 5H, H-14), 4.31–2.97 (m, 140H, PEG); ^13^C NMR (101 MHz, CDCl_3_) δ: 161.4 (C-2), 161.1 (C-7), 155.7 (C-8a), 143.4 (C-4), 142.7 (C-10 and C-13), 129.0 (C-5), 124.6 (C-11 and C-12), 113.4 (C-3), 112.9 (C-4a), 112.9 (C-6), 102.1 (C-8), 72.7 (PEG), 70.5 (PEG), 69.4 (PEG), 69.0 (PEG), 62.3 (C-9), 50.4 (C-14).

#### 3.3.26. α-3-[3-(4,8-Dimethyl]-7-(2-Propynyloxy)]-Coumarinyl Propanoate-ω-3-[3-(4,8-dimethyl]-7-(2-propynyloxy)]-coumarinyl propanoate PEG_1000_ (**26**)

Compound **3** (0.099 g, 0.3 mmol), DMAP (0.004 g, 0.03 mmol) and coupling agent DCC (0.075 g, 0.4 mmol) were stirred in DCM (7 mL) for 30 min. Then, PEG (0.150 g, 0.15 mmol) was added. After 16 h, DCM (10 mL) and acetic acid solution (0.25 M, 10 mL) were added to the reaction mixture. After 1 h the organic phase was separated and washed with deionized water (2 × 10 mL), dried over anhydrous sodium sulfate, filtered, evaporated and purified by silica gel column chromatography using chloroform/methanol 9:1, affording **26** (0.170 g, 74%) as a yellow wax. FT-IR (NaCl) ῡ_max_: 3259 (s, ≡C-H w), 3008 (s, C-H w), 2929 (s, C-H w), 1733 (s), 1701 (s, O-C=O st) cm^−1^; ^1^H NMR (400 MHz, CDCl_3_) δ: 7.47 (d, *J* = 8.9 Hz, 2H, H-5), 6.98 (d, *J* = 8.9 Hz, 2H, H-6), 4.82 (d, *J* = 2.2 Hz, 4H, H-1’), 4.23 (t, *J* = 4.6 Hz, 4H, H-16), 3.66 (s, 82H, CH_2_-PEG), 2.98 (t, *J* = 7.6 Hz, 4H, H-13), 2.66 (t, *J* = 7.6 Hz, 4H, H-14), 2.57 (t, *J* = 2.0 Hz, 2H, H-3’), 2.45 (s, 3H, H-12), 2.34 (s, 3H, H-11); ^13^C NMR (101 MHz, CDCl_3_) δ: 172.9 (C-15), 161.7 (C-2), 157.4(C-7), 151.4 (C-9), 147.5 (C-4), 122.5 (C-5), 121.7 (C-3), 115.0 (C-10), 114.7(C-8), 108.3 (C-6), 78.2 (C-2’), 77.3 (C-3’), 70.6 (CH_2_-PEG), 63.7 (C-16), 56.4 (C-1’), 32.6 (C-14), 23.1 (C-13), 15.0 (C-12), 8.3 (C-11).

#### 3.3.27. α-Thymidinyl-5-((1,2,3-Triazol-1-yl)-3-[3-(4,8-Dimethyl)-7-(Methyloxi)-Cumarinyl)propanoate-ω- Thymidinyl-5-((1,2,3-Triazol-1-yl)-3-[3-(4,8-Dimethyl)-7-(Methyloxi)-Cumarinyl)propanoate PEG_1000_ (**27**)

Compound **26** (0.060 g, 0.04 mmol), DIPEA (0.2 mL, mmol), copper iodide (0.008 g, cat.) and compound **16** (0.499 g, 1.3 mmol) were stirred in THF (4.0 mL) for 3 h. After solvent evaporation the mixture was dissolved in dichloromethane and purified by column chromatography using hexane/ethyl acetate (1:1) as eluent, affording **27** (0.070 g, 87%) as a yellow wax. ^1^H-NMR (CDCl_3_, 400 MHz) δ: 7.77 (d, *J* = 9.6 Hz, 2H, H-5’), 7.41 (d, *J* = 8.8 Hz, 2H, H-5), 7.30 (s, 2H, H-6’’), 6.98 (d, *J* = 8.7 Hz, 2H, H-6), 6.24 (t, *J* = 6.8 Hz, 2H, H-1’’), 4.70 (dd, *J* = 15.2, 10.0 Hz, 4H, H-6’), 4.47 (m, 2H, H-3’’), 4.20 (t, *J* = 5.1 Hz, 4H, H-16,), 4.02 (q, *J* = 3.9 Hz, 2H, H-4’’), 3.60 (s, *J* = 12.8 Hz, 100H, H-PEG, 5’’), 2.95 (t, *J* = 7,1 Hz, 4H, H-13), 2.62 (t, *J* = 7.0 Hz, 4H, H-14), 2.40–2.25 (m, 16H, H-11,12, 2’’), 1.92–1.86 (m, 6H, H-8’’); ^13^C-NMR (CDCl_3_, 100 MHz) δ: 172.9 (C-15), 163.8 (C-9’’), 161.9 (C-2), 158.1 (C-7), 151,3 (C-9), 150.4 (C-10’’), 147.9 (C-4), 143.8 (C-4’), 135.5 (C-5’), 124.8 (C-6’’), 122.8 (C-6), 121.3 (C-3), 114.7 (C-10), 114.1 (C-8), 111.4 (C-7’’), 108.3 (C-5), 84.9 (C-1’’), 84.5 (C-4’’), 71.5 (C-3’’), 71.2 (C-3’’), 70.5 (C-PEG), 63.7 (C-16), 52.3 (C-5’’), 51.3 (C-6’), 40.2 (C-2’’), 32.6 (C-14), 23.1 (C-13), 15.0 (C-12), 12.6 (C-8’’), 12.4 (C-8’’), 8.3 (C-11).

#### 3.3.28. 1-[1’-Ethylamide-Triazolyl-4]-1-O-Methyl-D-Glucopyranoside-PLGA Conjugate (**28**)

Compound **14** (0.006 g, 0.02 mmol) was dissolved in 4 mL of DMF, and commercially available polymer poly(d,l-lactide-co-glycolide) (0.200 g, 0.02 mmol) was added. To the stirred reaction mixture, methanesulfonic acid (0.5 mL, 7.70 mmol) was added and the reaction proceeded overnight at 65 °C. The solution was cooled down, and 4 mL of cold water was added. The precipitated polymer was collected by filtration to give the PLGA derivative **28** (0.193 g, 100%) as a white solid.

#### 3.3.29. 7-((1-(2-Amide-Ethyl)-1H-1,2,3-Triazol-4-yl)methoxy)-4-Methyl-2H-Chromen-2-One-PLGA Conjugate (**29**)

Compound **8** (0.006 g, 0.02 mmol) was dissolved in 4 mL of DMF, and poly(d,l-lactide-co-glycolide) (0.200 g, 0.02 mmol) was added and this mixture stirred at room temperature. Methanesulfonic acid (0.5 mL, 7.70 mmol) was added, and heated overnight at 65 °C. The solution was cooled and 4 mL of cold water was added. The precipitated polymer was collected by filtration to give the PLGA derivative **29** (0.197 g, 100%) as a white solid.

## 4. Conclusions

The use of these two polymeric building blocks (PEG and PLGA) has allowed us to prepare a new type of macromolecule functionalized with carbohydrate and coumarin moieties. The polymer PEG was selectively functionalized making possible the production of asymmetric macromolecules that contain at one terminal the carbohydrate unit and at the other the coumarin molecule. This asymmetry provides the ability to obtain a self-assembling control during the nanoparticles oil in water transformation. This kind of macromolecules showed a tendency to form aggregates with sizes ranging between 220 and 580 nm, respectively, but the majority of the sample presented a polymeric film after the medium evaporation. This film formation can suggest that the hydrophobic core is not big enough to provide the desired self-assembling effect. 

PLGA has been successfully functionalized and transformed into stable and spherical nanoparticles with a smooth surface, with practically no aggregation. The size of these different particles ranges between 114–289 nm with a zeta potential value of −28.2 mV for the glucoconjugate and −56.0 mV for the coumarin-containing derivative. Using a single oil in water emulsion technique, it was possible to obtain low polydispersity indexes for all PLGA nanoparticles.

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
