# Peer review of "Nanoparticles Based on Novel Carbohydrate-Functionalized Polymers"

_molecules, 2020, doi:10.3390/molecules25071744_

Round 1

Reviewer 1 Report

This manuscript describes the preparation of functionalized PLGA and PEG with coumarin and carbohydrate moieties following various synthetic routes. Subsequently, nanoparticles based on these derivatives were prepared using an oil-in-water emulsion method. This work may be publishable in Molecules after major revisions, since at several points should be reconsidered before acceptance to this journal.

Specifically:

  1. The results and discussion section should be rewritten presenting not only details about the followed synthetic routes but also proofs for the successful preparation of the products based on the characterization data (NMR, FTIR, MS). Full assignment of the NMR and FTIR peaks and discussion should be added.
  2. The authors mentioned that the PEG derivatives 20, 24 and 25 can form nanoparticles with sizes ranging from 33 to 60 nm or 19 to 54 nm, which are spherical with a smooth surface and also after drying, can form films. But from Micrograph 1 cannot be concluded something like that. Only agglomerates/precipitations are observed. DLS results and z-potential measurements should be added as in case of PLGA derivatives.
  3. In the title, the authors stated that these functionalized polymers could be used for targeted therapies but they did not present any experiment to justify this suggestion. If they want to keep this title at least in vitro experiment should be done otherwise the title should be modified.
  4. Minor comments:
  5. There are several type- errors that should be corrected
  6. Scheme 2, R2 and R4 of compound 1 are CH3
  7. A CH3 group is missing in the position 4 of compounds 1 and 29
  8. Scheme 7, in the cycloaddition reaction, the compound 10 was used not the compound 18 as was written.
  9. Correct (translate in English) the titles of the axis in figure 1.

Author Response

Dear reviewer,

First of all, I would like to thank you for all your suggestions.

All changes made are indicated below and duly marked.

Reviewer 1

This manuscript describes the preparation of functionalized PLGA and PEG with coumarin and carbohydrate moieties following various synthetic routes. Subsequently, nanoparticles based on these derivatives were prepared using an oil-in-water emulsion method. This work may be publishable in Molecules after major revisions, since at several points should be reconsidered before acceptance to this journal.

Specifically:

  1. “The results and discussion section should be rewritten presenting not only details about the followed synthetic routes but also proofs for the successful preparation of the products based on the characterization data (NMR, FTIR, MS). Full assignment of the NMR and FTIR peaks and discussion should be added.”

Revised:

Spectra discussion has been added.

Pages 2 and 3; lines 84 – 93

Page 3; lines 111 – 121

Page 5; lines 151 – 177

Page 5 and 6; lines 180 – 200

Page 6; lines 203 – 219

Page 7; lines 237

Page 7 and 8; lines 242 – 252

Page 8; lines 258 – 271

All the assignments were made to the compounds description, in section 3 (Materials and Methods).

Page 11; lines 337 – 343

Page 12; lines 351 – 356

Page 12; lines 362 – 368

Page 12; lines 377 – 383

Pages 12 and 13; lines 392 – 398

Page 13; lines 406 – 411

Page 13; lines 420 – 422

Page 13; lines 428 – 433

Page 14; lines 450 – 454

Page 14; lines 461 – 464

Page 14; lines 469 – 474

Page 14; lines 480 – 486

Page 15; lines 500 – 506

Page 15; lines 517 – 522

Page 15; lines 531 – 536

Page 16; lines 542 – 548

Page 16; lines 555 – 560

Page 16; lines 568 – 574

Page 16; lines 580 – 584

Page 17; lines 592 – 593

Page 17; lines 601 – 608

Page 17; lines 618 – 624

Page 18; lines 634 – 640

Page 18; lines 648 – 655

Page 18; lines 661 – 670

  1. “The authors mentioned that the PEG derivatives 20, 24 and 25 can form nanoparticles with sizes ranging from 33 to 60 nm or 19 to 54 nm, which are spherical with a smooth surface and also after drying, can form films. But from Micrograph 1 cannot be concluded something like that. Only agglomerates/precipitations are observed. DLS results and z-potential measurements should be added as in case of PLGA derivatives.”

Revised

PEG SEM analysis was altered according to Reviewer 1 advice, in a way that only agglomerates and films were observed. DLS results for these derivatives were added and discussed.

Page 8; lines 273 – 275

  1. “In the title, the authors stated that these functionalized polymers could be used for targeted therapies but they did not present any experiments to justify this suggestion. If they want to keep this title at least in vitro experiment should be done otherwise the title should be modified.”

Revised

We have changed the title because at the moment it is not possible to present the results of biological tests. This work is done by another research group, which, due to the global health situation, was forced to interrupt the work in progress for an indefinite time. It is our intention in the future, when possible, to invest in this scientific aspect.

In view of this reality, we opted to publish the synthetic part, as we consider that methods of synthesis of various versatile compounds are described, which could have application in several areas, thus being a useful chemical tool in different areas.

New title:

Nanoparticles based on novel carbohydrate-functionalized polymers

Page 1 – lines 2 and 3

  1. Minor comments:
  1. “There are several type- errors that should be corrected”

            Revised

           The typing errors have been corrected throughout the manuscript.

Page 1 – line 5

Page 1 – line 7

Page 1 – line 12

Page 1 – line 41

Page 2 – line 51

Page 2 – line 55

Page 2 – line 58

Page 2 – line 77

Page 3 – line 102

Page 3 – line 108

Page 4 – line 132

Page 4 – line 133

Page 11 – line 308

Page 11 – line 327

Page 11 – line 308

Page 11 – line 335

Page 11 – line 308

Page 11 – line 336

Page 11 – line 349

Page 12 – line 350

Page 12 – line 361

Page 12 – line 386

Page 12 – line 392

Page 13 – line 403

Page 13 – line 404

Page 14 – line 461

Page 14 – line 477

Page 15 – line 496

Page 15 – line 499

Page 15 – line 508

Page 15 – line 513

Page 15 – line 524

Page 16 – line 538

Page 16 – line 551

Page 16 – line 576

Page 18 – line 644

Page 18 – line 645

Page 18 – line 676

  1. Scheme 2, R2and R4of compound 1 are CH3

            Revised

Scheme 2 was corrected according to the reviewer’s suggestion, and also for the product 2 was added R2,R4=CH3.

Page 3; Scheme 2

  1. “A CHgroup is missing in the position 4 of compounds 1 and 29”

      Corrected

      We apologize for the mistake made in scheme 3 (it is not reagent 1, but 6), which created    confusion in the interpretation of the structures of compounds 1 and 29, which are correct.

Page 3; line 102

Page 3; Scheme 3

  1. “Scheme 7, in the cycloaddition reaction, the compound 10 was used not the compound 18 as was written.”

      Corrected.

Page 4; Scheme 5

  1. Correct (translate in English) the titles of the axis in figure 1.”

      Corrected

Both the X and Y axes in figure 1 were translated into English.

Page 10; Figure 2

Reviewer 2 Report

This manuscript reported that poly(ethylene glycol) (PEG) and poly(lactic-co-glycolic acid) (PLGA) were functionalized with coumarin and carbohydrate moieties such as thymidine, glucose, galactose, xylose and mannose that have high biological specificities via CuAAC “click” reaction. These functionalized polymers were used to obtain drug delivery agents, either por-drugs or nanoparticles for drug encapsulation. The synthetic work is huge and many products are obtained. I think this article can be accepted after fully addressing the following major issues:

  1. Although the authors demonstrate that those particles can load a variety of drugs on the surface, no biological evaluation data was provided. I think the authors should provide at least one bio-related result in the paper.
  2. The chapter number of the paper should be re-checked again.
  3. Perhaps a schematic illustration of ester hydrolysis of 2 to afford 3 should be given in the article?
  4. In this paper, there is no analysis of the MALDI MS spectra results, maybe the characteristic mass-charge ratio peak can be explained?
  5. The authors could add the following references which would again increase the interest to general “click” chemistry readers:Journal of Controlled Release 2018,273, 160-179; Polymer, 2017, 125, 303-329; Polym. Chem., 2019,10, 3806-3821; Coordination Chemistry Reviews, 2019, 380, 484-518.

Author Response

Dear reviewer,

First of all, I would like to thank you for all your suggestions.

All the changes made are indicated below and duly marked.

Reviewer 2

This manuscript reported that poly(ethylene glycol) (PEG) and poly(lactic-co-glycolic acid) (PLGA) were functionalized with coumarin and carbohydrate moieties such as thymidine, glucose, galactose, xylose and mannose that have high biological specificities via CuAAC “click” reaction. These functionalized polymers were used to obtain drug delivery agents, either por-drugs or nanoparticles for drug encapsulation. The synthetic work is huge and many products are obtained. I think this article can be accepted after fully addressing the following major issues:

  1. “Although the authors demonstrate that those particles can load a variety of drugs on the surface, no biological evaluation data was provided. I think the authors should provide at least one bio-related result in the paper.”

We changed the title because at the moment it is not possible to present the results of biological tests. This work is done by another research group, which, due to the global health situation, was forced to interrupt the work in progress for an indefinite time. It is our intention in the future, when possible, to invest in this scientific aspect.

In view of this reality, we opted to publish the synthetic part, as we consider that methods of synthesis of various versatile compounds are described, which could have application in several areas, thus being a useful chemical tool in different areas.

Page 1 – lines 2 and 3

  1. “The chapter number of the paper should be re-checked again.”

Corrected

The Section 3 was corrected with subsections (3.1, 3.2, etc.) and the Conclusion section was changed from 5 to 4.

Page 11; line 315

Page 11; line 326

Page 11; line 331

Page 11; line 332

Page 11; line 344

Page 12; line 357

Page 12; line 369

Page 12; line 384

Page 13; line 399

Page 13; line 412

Page 13; line 423

Page 13; line 434

Page 14; line 443

Page 14; line 455

Page 14; line 465

Page 14; line 475

Page 14; line 487

Page 15; line 494

Page 15; line 507

Page 15; line 523

Page 15; line 537

Page 16; line 549

Page 16; line 561

Page 16; line 575

Page 16; line 585

Page 17; line 594

Page 17; line 609

Page 17; line 625

Page 18; line 641

Page 18; line 656

Page 18; line 671

Page 19; line 677

Page 19; line 684

  1. “Perhaps a schematic illustration of ester hydrolysis of 2 to afford 3 should be given in the article? “

            Revised

            According the reviewer’s suggestion an explanatory note has been added at scheme 2.

Page 3; Scheme 2

  1. “In this paper, there is no analysis of the MALDI MS spectra results, maybe the characteristic mass-charge ratio peak can be explained?”

            Revised

MALDI-TOF results of PEG derivatives were added and discussed.

Page 5; lines 176 – 177

Page 6; lines 198 – 200

Page 6; lines 217 – 219

Page 8; lines 249 – 250

  1. “The authors could add the following references which would again increase the interest to general “click” chemistry readers:Journal of Controlled Release 2018,273, 160-179; Polymer, 2017, 125, 303-329; Polym. Chem., 2019,10, 3806-3821; Coordination Chemistry Reviews, 2019, 380, 484-518.”

            Revised

We appreciate the reviewer’s literature suggestions. We believe that this information contributes to the enrichment of our manuscript. We have included Journal of Controlled Release, 2018,273, 160-179 reference, due to the interesting review of the authors about cycloaddition reactions with polymers. In addition, Polym. Chem., 2019, 10, 3806-3821 was also added due to click chemistry with commercial available polymers. These articles are cited in section 1. Introduction.

Page 2; lines 66 – 68

Round 2

Reviewer 1 Report

The authors were taken under consideration all my suggestion and now I think that the manuscript is suitable for publication.

Reviewer 2 Report

I think the current revised version seems OK for me